# Fake News Classification in Urdu: A Domain Adaptation Approach for a Low-Resource Language

## Abstract

Misinformation on social media is a widely acknowledged issue, and researchers worldwide are actively engaged in its detection. However, low-resource languages such as Urdu have received limited attention in this domain. An obvious approach is to utilize a multilingual pretrained language model and fine-tune it for a downstream classification task, such as misinformation detection. However, these models struggle with domain-specific terms, leading to suboptimal performance. To address this, we investigate the effectiveness of domain adaptation before fine-tuning for fake news classification in Urdu, employing a staged training approach to optimize model generalization. We evaluate two widely used multilingual models, XLM-RoBERTa and mBERT, and apply domain-adaptive pretraining using a publicly available Urdu news corpus. Experiments on four publicly available Urdu fake news datasets show that domain-adapted XLM-R generally outperforms its vanilla counterpart, while domain-adapted mBERT exhibits mixed results. These findings highlight the varying impact of domain adaptation across multilingual architectures in low-resource settings. We release our domain-adapted models, code, and datasets at URL withheld.

## 1 Introduction

Fake news refers to any statement or piece of text that is verifiably false and contradicts established truths Allcott & Gentzkow (2017). While the author's intention behind such content is often debated, its societal impact is undeniable. Although fake news has existed for centuries, its prevalence has dramatically increased due to the rise of the internet and social media. This proliferation is largely driven by the public's tendency to share unverified information online, leading to adverse effects, including influencing public opinion, undermining trust in media, and even altering election outcomes Jang & Kim (2018). Studies have shown that fake news spreads significantly faster than truthful information, making its detection an important area of research Langin (2018).

To tackle the challenge of misinformation detection, researchers have explored diverse feature sets, including *content*, *social*, *temporal*, and *hybrid* Shu et al. (2019), as well as a wide range of learning architectures, from traditional models like decision trees Krishna & Adimoolam (2022) to deep learning methods such as LSTMs and RNNs Mridha et al. (2021), and more recently, large pretrained language models Kaliyar et al. (2021). These features and architectures are discussed in detail in the Related Works section.

Despite extensive research on fake news detection for resource-rich languages like English Alghamdi et al. (2024), studies focusing on Urdu, a low-resource language, remain scarce Ahmed et al. (2017); Previti et al. (2020). Consequently, there is a lack of both datasets and automated methods for detecting fake news in this language Amjad et al. (2020). Detecting fake news in Urdu is particularly important due to its approximately 232 million speakers International Center for Language Studies (n.d.), its status as the national language of Pakistan[1], and its growing use in social media communication Ullah et al. (2024). This widespread use increases the risk of people believing and sharing fake news, highlighting the pressing need for automated detection solutions for Urdu. Existing efforts to detect fake news in Urdu often simplify the problem as text classification, relying on surface-level linguistic features, such as vocabulary usage, writing

---

[1] https://en.wikipedia.org/wiki/Urdu

styles and sentiment Saeed et al. (2021); Kausar et al. (2020). The advent of pretrained language models has made it feasible to leverage complex hidden features within fake news datasets, enabling the development of more sophisticated classifiers Shehata et al. (2024); Ibrahim Aboulola & Umer (2024). However, there is a notable absence of Urdu-specific pretrained language models. While some multilingual models can process Urdu, they are predominantly trained on resource-rich languages like English, with Urdu constituting only a small fraction of their training data Carlsson et al. (2022); Conneau et al. (2019). As a result, these multilingual models often struggle to understand Urdu-specific terms and nuances. Moreover, these models face challenges in specific domains due to their general-purpose training on diverse, non-specific datasets, a phenomenon referred to as *domain shift* in NLP and *dataset shift* in machine learning, where the training and test distributions do not align Gretton et al. (2006). Fine-tuning such models on domain-specific datasets can lead to domain shifts, where the model struggles with *out-of-distribution generalization*, resulting in suboptimal performance Glorot et al. (2011).

To address these problems, we perform domain adaptation on two widely used multilingual pretrained language models, XLM-RoBERTa (XLM-R) Conneau et al. (2019) and mBERT Devlin et al. (2018), using a publicly available corpus of one million Urdu news articles. This approach offers several advantages: (i) it enables the model to update its weights for domain-specific terms and learn their contextual usage, and (ii) it enhances their handling of Urdu vocabulary, which is often underrepresented in multilingual pretraining. Additionally, by exposing the models to authentic Urdu news content during the domain adaptation phase, we aim to help them learn latent domain-specific features that may improve performance in downstream tasks such as fake news detection.

To evaluate the impact of domain adaptation, we fine-tune both adapted and vanilla versions of XLM-R and mBERT on four publicly available Urdu fake news datasets. Our results show that the domain-adapted XLM-R consistently outperforms its vanilla counterpart across all datasets. In contrast, mBERT shows mixed results, with only marginal gains in some cases and no improvement or slight degradation in others. These findings highlight the potential of domain adaptation for improving fake news detection in low-resource languages, while also emphasizing the importance of model choice in this setting.

## 2 Related Works

This section reviews previous work on fake news detection, including studies focused on Urdu, as well as prior research on domain adaptation.

**Fake News Detection**  Existing works on fake news detection can be categorized by the features used to train classifiers and the learning architectures employed Alghamdi et al. (2024). **Feature sets** include content, social, temporal, or their combinations. *Content features* rely on the use of linguistic cues such as grammar, vocabulary, and sentiment Reis et al. (2019). *Social Features* leverage post-related meta-information and user-data from social media, such as post-likes, user followers, and followed pages Castillo et al. (2011). *Temporal Features* use time-based heuristics extracted from post timestamps Kwon et al. (2017). **Classifier architectures** have evolved from simple machine learning models Krishna & Adimoolam (2022) like Decision Trees to advanced deep learning architectures including LSTMs and RNNs Mridha et al. (2021), and more recently to fine-tuned pretrained language models and LLMs Kaliyar et al. (2021). For instance, Dhiman et al. (2024) proposed a Generative BERT framework combining GPT and BERT features for fake news detection, reporting improved results on two public datasets. Recent studies have focused on detecting AI-generated fake news: Stewart et al. (2023) fine-tuned BERT to identify GPT-generated articles, while Su et al. (2023); Ali et al. (2025) developed methods that both detect fake news and identify machine-written content, highlighting the need for new labeling approaches.

Urdu Fake News Detection remains relatively under-explored, both in feature selection and model architectures. Farooq et al. (2023) presented an Urdu Fake news dataset with 4097 headlines across 9 domains but used simple feature sets (TF-IDF, bag-of-words) with models like SVM, Random Forest, and Extra Trees. Similarly, Iqbal et al. (2024) presented an Urdu fake news dataset comprising 12k news articles, and trained classifiers using machine learning models (SVM, Random Forest) and deep learning models (CNN, RNN), comparing their performance. Recent work by Hussain et al. (2025) employed attention mechanisms

on COVID-19 data, reporting high accuracy. However, most studies treat Urdu fake news detection as a basic text classification task Akhter et al. (2021); Kausar et al. (2020); Amjad et al. (2020), showing limited methodological diversity.

**Domain Adaptation** Domain adaptation is a technique used to mitigate performance degradation caused by domain shift– the difference in data distributions between the source domain (training data) and the target domain (test data) Volpi et al. (2018). This challenge is particularly prevalent in natural language processing (NLP), where variations in language style, vocabulary, and context can largely impact the model's performance. The two main approaches are: *supervised domain adaptation* Daumé III (2009), requiring labeled target domain data, and *unsupervised domain adaptation* Ramponi & Plank (2020), employing techniques like adversarial training Yasunaga et al. (2017), self-training Yu et al. (2015), and contrastive learning Wang et al. (2022). Our work focuses on unsupervised domain adaptation due to limited labeled Urdu fake news data Akhter et al. (2021). Various methods have been employed for unsupervised domain adaptation, including domain-invariant representation learning via adversarial techniques Ganin & Lempitsky (2015), and self-supervised objectives like masked language modeling and contrastive learning for alignment Gururangan et al. (2020). In domain adaptation for fake news detection, recent studies have explored approaches to address linguistic and contextual differences across datasets. Ozcelik et al. (2023) explored cross-lingual transfer learning for misinformation detection across five languages, evaluating model performance across language boundaries. Similarly, Huang et al. (2021) proposed a domain adaptation framework that enhances fake news detection when encountering unseen domains during inference. Their dual strategy combines: (i) distribution alignment during pre-training, and (ii) adversarial example generation in embedding space during fine-tuning. Li et al. (2021) introduced a multi-source adaptation approach using weak supervision to improve early detection by employing multiple domains.

Our approach presents the first study on domain adaptation for Urdu fake news detection. We adapt the masked language models of both XLM-R and mBERT using Urdu news articles, then fine-tune them on publicly available Urdu fake news datasets, comparing their performance against their respective vanilla baselines.

## 3 Methodology

This section provides an overview of the datasets utilized in our study and outlines the proposed architecture employed in our experiments.

### 3.1 Datasets

We use five publicly available Urdu datasets in this work: one for domain adaptation and four labeled fake news datasets for downstream classification tasks.

#### 3.1.1 Urdu News Articles Dataset

*Urdu 1M Dataset* Hussain et al. (2021) is used for domain adaptation of the pretrained XLM-R and mBERT models. It comprises 1 million Urdu news articles across four categories: Business & Economics, Science & Technology, Entertainment, and Sports. Table 1 shows the exact number of examples per category. About 70% of the articles are sourced from Pakistan's leading news outlets, Geo News and Dawn News, while the rest come from other PEMRA[2]-regulated media houses.

#### 3.1.2 Fake News Datasets

Four publicly available Urdu fake news datasets are used for fine-tuning a pretrained language model on the downstream classification task: *Ax-to-Grind Urdu* Harris et al. (2023), *UFN2023* Farooq et al. (2023), *UrduFake2021* Amjad et al. (2020), and *UFN2021* Akhter et al. (2021). The first two contain short headlines, while the latter two consist of longer news articles. Real news in the first three datasets was collected from

---

[2]https://pemra.gov.pk/

Table 1: Summary of datasets used in training. Labels: $S$ = Sports, $E$ = Entertainment, $B$ = Business, $T$ = Technology, $F$ = Fake, $R$ = Real.

| Dataset | #Examples | Labels | Content | Application |
|---------|-----------|--------|---------|-------------|
| *Urdu 1M Dataset* Hussain et al. (2021) | 1,038,340 | $S$: 454,335
$E$: 253,730
$B$: 251,170
$T$: 79,105 | Articles | Domain Adaptation |
| *Ax-to-Grind* Harris et al. (2023) | 10,083 | $F$: 5,053
$R$: 5,030 | Headlines | Classifier Training |
| *UFN2023* Farooq et al. (2023) | 4,097 | $F$: 2,455
$R$: 1,642 | Headlines | |
| *UFN2021* Akhter et al. (2021) | 2,000 | $F$: 968
$R$: 1,032 | Articles | |
| *UrduFake2021* Amjad et al. (2020) | 1,300 | $F$: 550
$R$: 750 | Articles | |

credible PEMRA-regulated sources, whereas fake news was gathered from controversial websites or through crowd-sourcing. In *UrduFake2021*, journalists were hired to create fake news by crafting counterfactual narratives of real articles in a journalistic tone. The *UFN2021* Akhter et al. (2021) differs as it was generated by machine-translating English fake news with human supervision. Table 1 summarizes the complete dataset statistics.

## 3.2 Proposed Architecture

Figure 1 shows the proposed architecture of our study, where we investigate the performance of two multi-lingual pretrained language models, XLM-R and mBERT, for Urdu fake news detection. XLM-R is adopted due to its strong cross-lingual performance and support for Urdu in its vocabulary Conneau et al. (2019); Hu et al. (2020), while mBERT is included as a widely used baseline for multilingual NLP Devlin et al. (2018). To assess the impact of domain adaptation, we compare two approaches for each model:

- **Basic fine-tuning** (left side): The Pretrained Language Model (PLM) is directly fine-tuned on the Urdu Fake News Detection dataset, resulting in PLM$_{\text{FND}}$. This represents the standard application of the model to the task.

- **Domain adaptation + fine-tuning** (right side): PLM first undergoes domain adaptation, tailoring it to the Urdu news domain and producing PLM$_{\text{UN}}$. This adapted model is then fine-tuned on the Urdu Fake News Detection dataset, yielding PLM$_{\text{UNFND}}$.

### 3.2.1 Domain Adaptation Process

For unsupervised domain adaptation, we apply masked language model (MLM) fine-tuning to XLM-R and mBERT using a corpus of 1 million Urdu news articles. The dataset was split into training (80%), validation (10%), and test (10%) sets. During preprocessing, all examples in the dataset were concatenated and split into equal-sized chunks of 128. This size was chosen due to GPU memory constraints, to prevent truncation of long examples, and to preserve valuable information for language modeling. The last chunk was smaller than the specified size and was consequently dropped to maintain uniformity. During fine-tuning, 15% of the tokens in each batch are randomly masked, following the standard setup for BERT-family models Devlin et al. (2018); Liu et al. (2019). Additionally, whole word masking is applied with a probability of 0.2, where

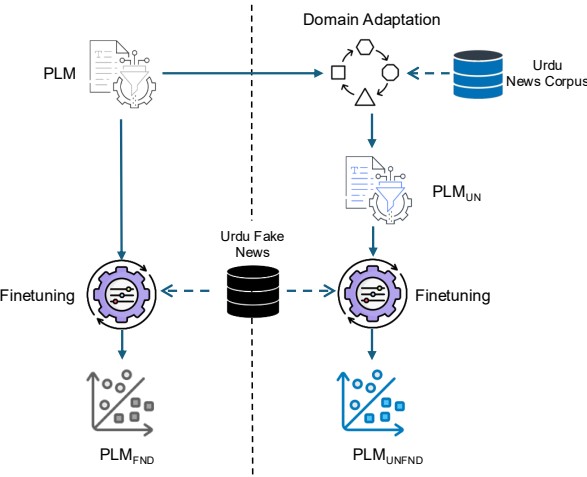

Figure 1: Proposed architecture. (left side) Basic fine-tuning; (right side) Domain adaptation + fine-tuning. **PLM** = Pretrained Language Model.

entire words, rather than subword tokens, are masked together. This helps the model better capture semantic and contextual information at the word level Cui et al. (2021); Joshi et al. (2020).

### 3.2.2   Fine-tuning the Model

Fine-tuning for the fake news classification task is performed by extending the pretrained language model with additional layers designed to stabilize learning and improve generalization. Initially, we experimented with minimal layers, such as using only a single dense layer followed by the output layer on top of the pretrained model. However, these setups resulted in unstable training curves, with oscillating loss values and poor convergence. To address this, we incorporated two dense layers (256 and 128 neurons, respectively), each followed by batch normalization, and a dropout layer (rate=0.5) before the final classification layer. The dense layers employ ReLU activation functions to introduce non-linearity, which enables the model to capture more complex patterns from the transformer representations Nair & Hinton (2010). Batch normalization stabilizes training by normalizing activations, mitigating internal covariate shifts, and improving convergence speed Ioffe & Szegedy (2015). The dropout layer randomly deactivates neurons during training, which is a well-established technique for regularization and helps prevent overfitting Srivastava et al. (2014). The combination of these components provided a balance between model capacity to capture complex patterns and training stability, which in turn resulted in smoother learning curves and more reliable convergence in our experiments. Since the task is binary classification, the final layer consists of a single neuron with a sigmoid activation function, mapping the learned representations to probability scores (0 = real, 1 = fake). As a preprocessing step, the labels in all four datasets were numerically encoded, and each dataset was split into 68% training, 12% validation, and 20% testing.

Training is conducted in two stages. In the first stage, only the layers added after the pretrained model are trained while the base model remains frozen for 20 epochs. This allows the model to adapt to task-specific representations while preventing excessive parameter updates, which has shown to reduce the risk of catastrophic forgetting in transfer learning scenarios Chen et al. (2020); Iman et al. (2023). Using the best-performing weights from this stage, the second stage involves unfreezing the entire model and fine-tuning it with a lower learning rate for another 20 epochs. This ensures that while the model refines its feature representations, the pretrained knowledge remains stable. The final model incorporates the best weights from both stages, leveraging both general pretrained representations and task-specific adaptations.

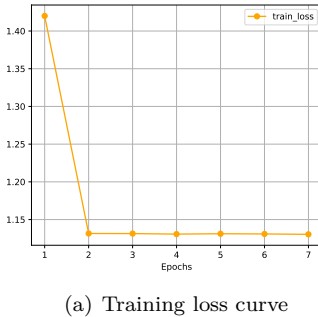
(a) Training loss curve

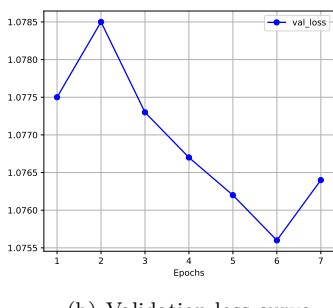
(b) Validation loss curve

Figure 2: Domain adaptation (Masked Language Model) loss curves for XLM-R model.

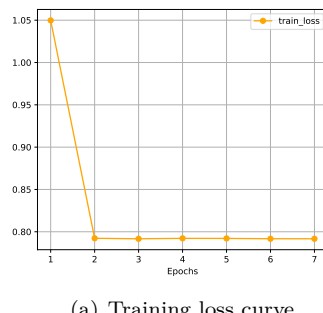
(a) Training loss curve

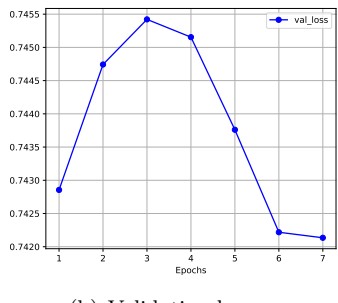
(b) Validation loss curve

Figure 3: Domain adaptation (Masked Language Model) loss curves for mBERT model.

## 4 Experimental Setup

This section details the model's training parameters, presents the loss and accuracy curves, and discusses the evaluation metrics used to assess performance.

### 4.1 Model Training

#### 4.1.1 Masked Language Model

The `TFAutoModelForMaskedLM` class from the HuggingFace Transformers library was used for this experiment. Both models, XLM-R and mBERT, were trained for 7 epochs with a batch size of 16 and an initial learning rate of $1 \times 10^{-4}$. The optimizer followed a linear decay schedule with 1,000 warm-up steps and a weight decay rate of 0.01. The number of training steps was set to the total number of batches in the training data for each model. These hyperparameter choices are aligned with best practices recommended in the Hugging Face documentation and tutorial on masked language modeling Hugging Face (2025). To further refine these settings, we conducted a series of empirical experiments, tweaking the hyperparameters to identify a well-performing configuration that yielded stable training curves and lower validation perplexity. Since the `TFAutoModelForMaskedLM` model is designed for masked language modeling, it inherently uses the masked language modeling loss, which computes cross-entropy only over the masked tokens. The training dataset was shuffled before each epoch, while the evaluation dataset remained unshuffled to maintain consistency.

**XLM-R model** The training and validation loss curves for the XLM-R model are shown in Figure 2. The training loss drops sharply between the first and second epoch, indicating that the model quickly learns basic patterns. After this initial drop, the loss stabilizes around 1.13, with only minor fluctuations. The validation

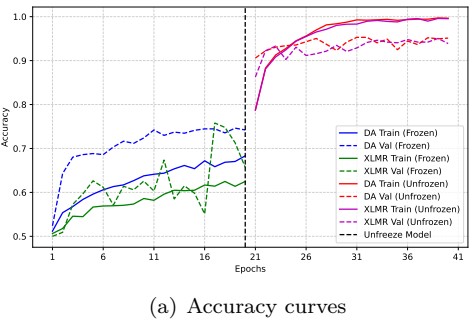 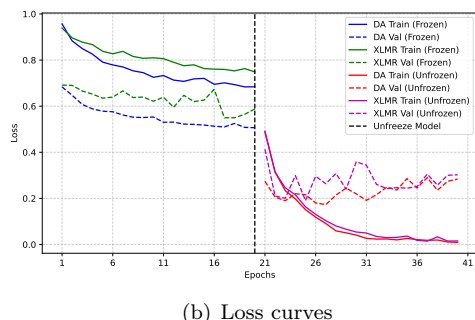

(a) Accuracy curves          (b) Loss curves

Figure 4: Accuracy and loss curves for classifier trained on *Ax-to-Grind* Harris et al. (2023) using XLM-R models.

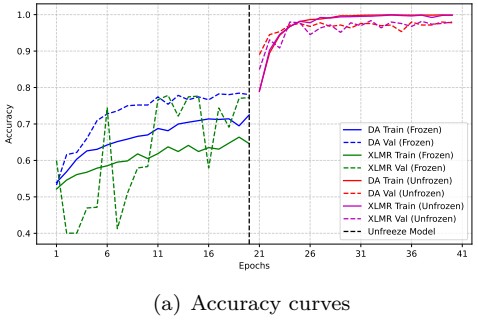 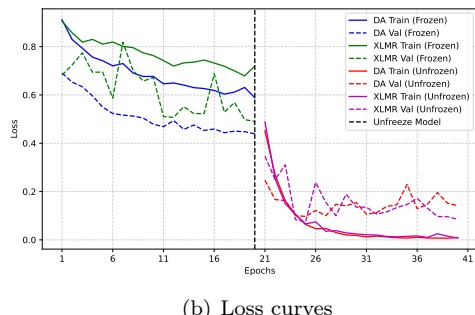

(a) Accuracy curves          (b) Loss curves

Figure 5: Accuracy and loss curves for classifier trained on *UFN2023* Farooq et al. (2023) using XLM-R models.

loss, on the other hand, shows a different trend: it increases slightly after the first epoch, gradually decreases until the sixth epoch, and then begins to rise again. However, these fluctuations are minimal (on the order of 0.0005), suggesting that the validation loss remains effectively stable after the first epoch. Throughout training, the validation loss consistently remains lower than the training loss, which is expected given the larger size and greater token diversity of the training set, leading to a higher training loss in masked language modeling.

**mBERT model**  Figure 3 shows the loss curves from the masked language modeling stage for mBERT. The training loss curve closely resembles that of XLM-R, with the only difference being lower overall loss values. Although the validation loss curve appears different from that of XLM-R at first glance, a closer look shows that the difference between successive epochs remains around 0.0005. This again suggests that the validation loss remains effectively stable throughout training.

### 4.1.2 Fake News Classifier

To fine-tune the language models for the downstream fake news classification task, binary cross-entropy was used as the loss function, with Adam as the optimizer, a batch size of 32, and 15% of training data as validation. Again, these hyperparameters were selected through empirical experiments, tweaking them to identify a well-performing configuration for this specific task. Training was conducted in two stages: initially, the pretrained model was frozen while the newly added layers were trainable, followed by unfreezing the entire model. A learning rate of $1 \times 10^{-5}$ was used during the frozen stage, while a smaller learning rate of $1 \times 10^{-6}$

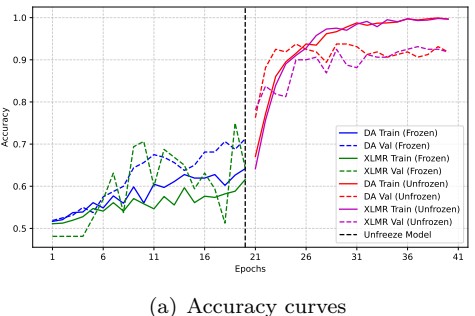
(a) Accuracy curves

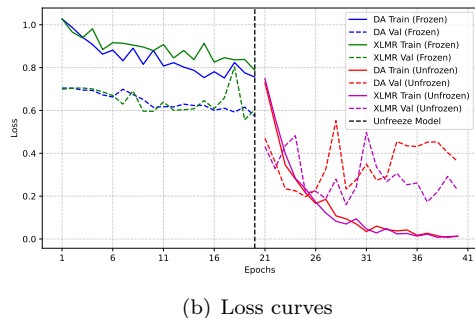
(b) Loss curves

Figure 6: Accuracy and loss curves for classifier trained on *UFN2021* Akhter et al. (2021) using XLM-R models.

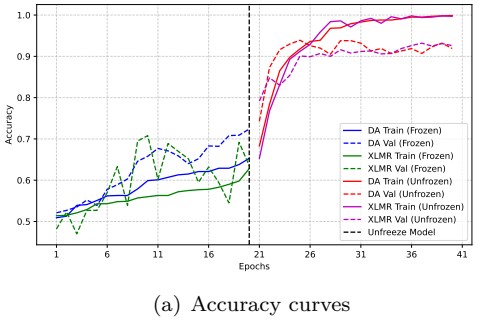
(a) Accuracy curves

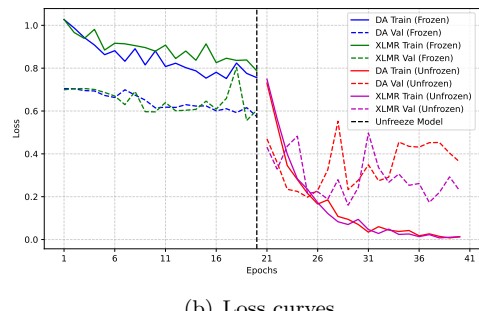
(b) Loss curves

Figure 7: Accuracy and loss curves for classifier trained on *UrduFake2021* Amjad et al. (2020) using XLM-R models.

was applied when the entire model was trainable. The reduced learning rate in the unfrozen stage prevents drastic weight updates, ensuring smoother convergence.

**XLM-R variants** The training and validation loss curves for all datasets using the domain-adapted (DA XLMR) and vanilla (XLMR) models are shown in Figures 4–7. Across all datasets, both models exhibit a general decrease in loss and an increase in accuracy over time, though their behaviors differ significantly in the frozen stage. The domain-adapted XLM-R model shows a steady increase in training accuracy with minimal variance across epochs, while its validation accuracy remains consistently higher than training accuracy. This suggests that the domain-adapted representations are already well-suited for the task, leading to better generalization even with limited fine-tuning. In contrast, vanilla XLM-R exhibits highly fluctuating validation accuracy during the frozen stage. While the accuracy improves over epochs, it does not follow a stable upward trajectory, indicating that the model struggles to learn meaningful task-specific features without domain adaptation. The instability is further reflected in the loss curve, which shows occasional increases in validation loss, suggesting difficulty in generalization at this stage. Comparing the learning curves of short datasets (Figures 4–5) and long datasets (Figures 6–7) during frozen stage, it can be noticed that the domain adapted XLM-R identifies the underlying patterns in the short datasets with relative ease. This is expected as it is difficult to find patterns that discern fake news from true news in long datasets due to less information density.

During the unfrozen stage, both models undergo significant improvements, with training accuracy exceeding 99% in later epochs. However, domain-adapted XLM-R generally reaches convergence slightly earlier than vanilla XLM-R. Additionally, a key shift occurs: while validation accuracy initially surpasses training

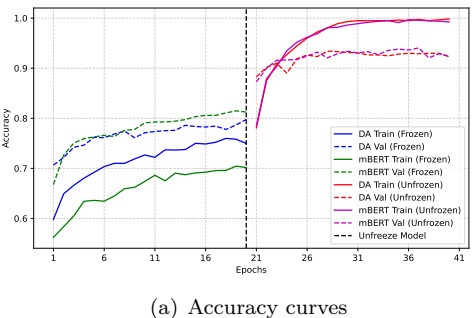

(a) Accuracy curves

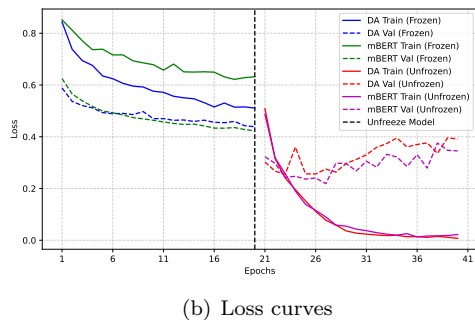

(b) Loss curves

Figure 8: Accuracy and loss curves for classifier trained on *Ax-to-Grind* Harris et al. (2023) using mBERT models.

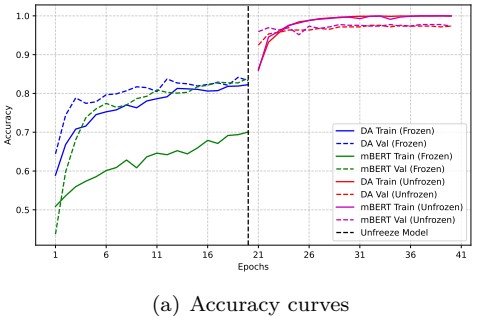

(a) Accuracy curves

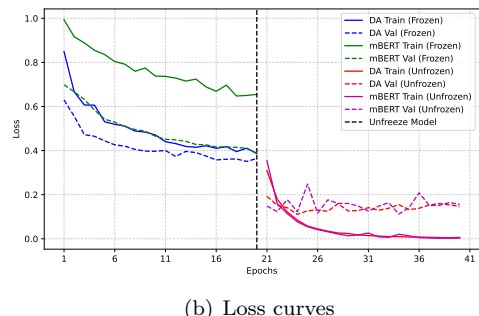

(b) Loss curves

Figure 9: Accuracy and loss curves for classifier trained on *UFN2023* Farooq et al. (2023) using mBERT models.

accuracy in the frozen stage, it eventually settles below training accuracy for the domain-adapted model. This suggests that as the model fully fine-tunes, it starts fitting more closely to the training data, capturing task-specific patterns. Although training accuracy continues to rise, validation accuracy plateaus, indicating that additional fine-tuning does not necessarily improve generalization beyond a certain point. Moreover, the increasing validation loss points to overfitting.

Overall, training loss and accuracy curves observed across all four datasets for XLM-R highlight the benefits of domain adaptation in stabilizing early learning and promoting faster convergence.

**mBERT variants**   Accuracy and loss curves for mBERT variants are shown in Figures 8–11. The training accuracy curve for the domain-adapted mBERT model is generally higher than its vanilla counterpart. Across all datasets, the validation accuracy curve is higher than the training curve, but the gap is smaller for the domain-adapted version, suggesting that the lower perplexity did not translate into improved learning during downstream fine-tuning, as was observed in the case of XLM-R. During the unfrozen stage, both variants perform similarly across all datasets. The learning curves for Dataset 4 appear notably flat, indicating that neither variant learned effectively during training. This may be due to the smaller size of the training set, which makes it difficult for mBERT models to generalize. This is also reflected in Table 3, where both mBERT models achieve accuracies in the 60s. In summary, mBERT exhibits a different pattern compared to XLM-R, particularly during the frozen stage, where both vanilla and domain-adapted variants show nearly identical behavior. This suggests that, unlike XLM-R, domain adaptation has a limited effect on mBERT during early training.

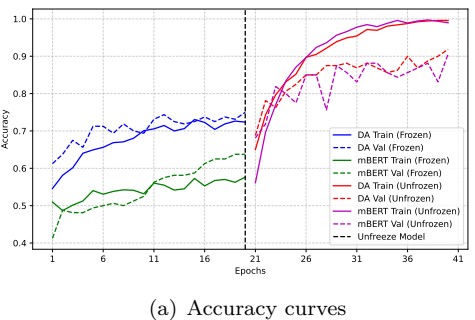
(a) Accuracy curves

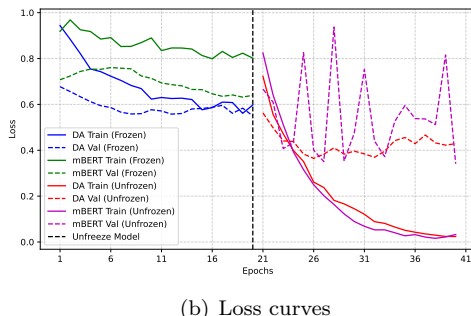
(b) Loss curves

Figure 10: Accuracy and loss curves for classifier trained on *UFN2021* Akhter et al. (2021) using mBERT models.

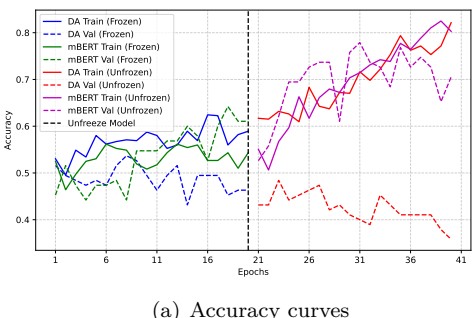
(a) Accuracy curves

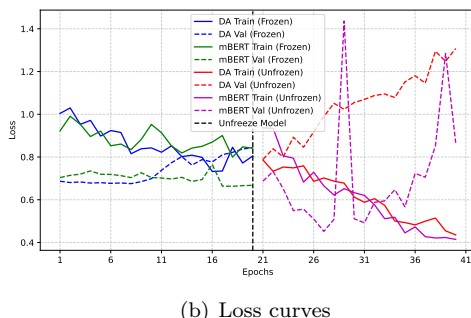
(b) Loss curves

Figure 11: Accuracy and loss curves for classifier trained on *UrduFake2021* Amjad et al. (2020) using mBERT models.

Table 2: Perplexity comparison of vanilla and domain-adapted models.

| Model | Perplexity |
|---|---|
| XLM-R (vanilla) | 11.11 |
| XLM-R (DA) | 3.33 |
| mBERT (vanilla) | 12.39 |
| mBERT (DA) | **2.10** |

## 4.2 Evaluation Metrics

To assess the performance of the masked language model, we used perplexity, an intrinsic evaluation metric that quantifies how well a probabilistic model predicts a given sequence. It is defined as the exponentiation of the average log-likelihood of the test set Martin & Jurafsky (2009), as shown in the following formula:

$$PPL = 2^{H(M)} \tag{1}$$

where $H(M)$ is the cross-entropy of the model, i.e., overall loss on the test set. Lower perplexity values indicate better predictive performance, as the model assigns higher probabilities to the correct sequences.

For evaluating the fake news classification models, accuracy, precision, recall, and F1-score are used. *Accuracy* measures the overall proportion of correct predictions across all classes. *Precision* reflects how many of the instances predicted as fake were actually fake, indicating the reliability of the model's positive predictions. *Recall* captures how many of the actual fake instances the model correctly identified. The *F1-score* balances

Table 3: Performance comparison across models on four Urdu fake news datasets: **ATG** = Ax-to-Grind, **UFN23** = UrduFakeNews 2023, **UFN21** = UrduFakeNews 2021, **UFake21** = UrduFake 2021. **Van** = Vanilla. **DA** = Domain-adapted.

| Dataset | Model | Precision | Recall | F1-score | Accuracy |
|---|---|---|---|---|---|
| ATG | LSVM | 0.90 | 0.87 | 0.88 | 0.89 |
| | mBERT (Van) | 0.91 | **0.94** | **0.93** | **0.93** |
| | mBERT (DA) | **0.94** | 0.91 | **0.93** | **0.93** |
| | XLM-R (Van) | 0.90 | 0.93 | 0.91 | 0.91 |
| | XLM-R (DA) | **0.94** | 0.92 | **0.93** | **0.93** |
| UFN23 | LSVM | 0.93 | 0.96 | 0.95 | 0.93 |
| | mBERT (Van) | 0.98 | 0.96 | 0.97 | 0.97 |
| | mBERT (DA) | 0.98 | 0.96 | 0.97 | 0.97 |
| | XLM-R (Van) | 0.98 | **0.97** | 0.97 | 0.97 |
| | XLM-R (DA) | **1.00** | 0.96 | **0.98** | **0.98** |
| UFN21 | LSVM | **0.93** | 0.88 | **0.91** | **0.91** |
| | mBERT (Van) | 0.86 | 0.93 | 0.89 | 0.89 |
| | mBERT (DA) | 0.88 | 0.85 | 0.87 | 0.87 |
| | XLM-R (Van) | 0.85 | **0.94** | 0.89 | 0.89 |
| | XLM-R (DA) | 0.90 | 0.91 | 0.90 | **0.91** |
| UFake21 | LSVM | 0.72 | 0.69 | 0.70 | 0.73 |
| | mBERT (Van) | 0.90 | 0.43 | 0.58 | 0.64 |
| | mBERT (DA) | 0.65 | 0.73 | 0.69 | 0.62 |
| | XLM-R (Van) | 0.83 | 0.81 | 0.82 | 0.84 |
| | XLM-R (DA) | **0.85** | **0.82** | **0.83** | **0.85** |

precision and recall, offering a more comprehensive measure of classification performance, particularly in cases of class imbalance.

## 5    Baseline Model

To evaluate the effectiveness of domain adaptation, we compare the performance of the vanilla and domain-adapted versions of each model using perplexity.

For the fake news classification task, we report the results of both the vanilla and domain-adapted versions of each model. Although prior works specify the models used and report results, they do not provide their code or exact test splits, making it difficult to replicate their experimental setup. As a result, direct comparison with their results is not feasible. The primary focus of this study is to assess the impact of domain adaptation by comparing the performance of vanilla and domain-adapted versions of both XLM-R and mBERT. Accordingly, the vanilla models are used as baselines. Additionally, a hyperparameter-tuned Linear SVM (LSVM) with TF-IDF features is included to provide a non-transformer-based comparison, highlighting the extent of improvement achieved by transformer models.

## 6    Results & Discussion

Table 2 presents the perplexity scores for the masked language models. In both cases, the domain-adapted models achieve significantly lower perplexity than their vanilla counterparts (3.33 vs. 11.11 for XLM-R, and 2.10 vs. 12.39 for mBERT), indicating that domain adaptation leads to more confident and coherent predictions.

Table 3 summarizes the performance of the fake news classification models across four datasets. In general, the performance of transformer models is significantly better than the hyperparameter-tuned LSVM model across all four datasets, except UFN2021 Akhter et al. (2021), where LSVM achieves performance comparable to the best transformer model. This suggests that lexical features in fake and true news are sufficiently distinct in this dataset, enabling a simple TF-IDF-based LSVM to perform competitively. This is expected, as the dataset is not organic but rather a machine-translated version of an English fake news dataset, as

mentioned in Section 3.1.2. However, the overall results indicate that deep contextual understanding, as provided by transformer-based models, is important for more robust and generalizable performance. Among the transformer models, XLM-R variants generally outperform mBERT across all datasets except Ax-to-Grind, where mBERT variants perform on par with domain-adapted XLM-R. This can be attributed to XLM-R's larger model size, better coverage of Urdu in its pretraining data, and dynamic masking during training.

**Headlines vs. Full Articles**  The models achieve higher scores on headline datasets (Ax-to-Grind Harris et al. (2023) and UFN2023 Farooq et al. (2023)) compared to long news article datasets (UFN2021 Akhter et al. (2021) and UrduFake2021 Amjad et al. (2020)). This can be attributed to: (i) the absence of information loss or truncation during tokenization for short headline datasets, and (ii) the higher information density in short headlines, which makes it easier for the models to learn meaningful patterns. In contrast, long articles often contain extraneous information, making classification more challenging.

**XLM-R (Domain-adapted vs. Vanilla)**  We observe a consistent improvement across all datasets. Notably, for the Ax-to-Grind dataset Harris et al. (2023), domain adaptation improves both accuracy (0.93 vs. 0.91) and F1-score (0.93 vs. 0.91). In the UFN2023 Farooq et al. (2023) dataset, the domain-adapted model achieves perfect precision, and a higher F1-score (0.98 vs. 0.97). In UFN2021 Akhter et al. (2021), both accuracy and F1-score show noticeable improvements, despite LSVM performing comparatively well. For the UrduFake2021 Amjad et al. (2020), while the absolute gains are smaller, the domain adaptation model still achieves higher accuracy (0.85 vs. 0.84) and F1-score (0.83 vs. 0.82).

**mBERT (Domain-adapted vs. Vanilla)**  The effects of domain adaptation on mBERT are less consistent. Despite showing a substantial drop in perplexity (Table 2), mBERT does not consistently benefit in downstream classification tasks. For instance, on Ax-to-Grind dataset Harris et al. (2023), both mBERT variants perform similarly (F1 = 0.93), with slight variation in precision and recall. On UFN2023 Farooq et al. (2023), domain adaptation does not yield further gains. Most notably, on UrduFake2021 Amjad et al. (2020), the domain-adapted mBERT model underperforms its vanilla version in accuracy (0.62 vs. 0.64) and F1-score (0.69 vs. 0.58). This disconnect between lower perplexity and downstream gains suggests that mBERT does not benefit from domain adaptation as effectively as XLM-R in this setting.

Overall, domain adaptation has a potential to improve fake news detection in low-resource setting, but its effectiveness depends heavily on the model. Not all models benefit equally; mBERT, for instance, shows lower perplexity but inconsistent downstream gains. This indicates that reducing perplexity does not necessarily translate to better classification performance. Model choice is therefore critical for making domain adaptation work in practice.

## 7    Conclusion & Future Dimension

This study investigates the impact of domain adaptation on multilingual models for fake news detection in Urdu. While both XLM-R and mBERT show lower perplexity after adaptation, only XLM-R consistently translates this into improved classification performance. These findings indicate that, although domain adaptation can be beneficial, its effectiveness depends strongly on the model architecture. Reducing perplexity alone is not a reliable indicator of downstream success in low-resource settings. Limitations and broader impact are discussed in the appendix.

For future work, incorporating additional datasets can further enhance domain adaptation and enable a broader evaluation of its impact. Exploring its applicability in other domains with more general Urdu corpora may provide insights into its versatility. Research into advanced adaptation techniques, such as continual learning and prompt tuning, may further refine model performance. Finally, investigating the model's robustness against adversarial attacks and misinformation generated by large language models, particularly for low-resource languages, remains a highly promising direction to ensure trust and reliability in real-world applications.

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

## A   Appendix

## Limitations

Our approach relies on publicly available datasets for both domain-adaptive pretraining and downstream fine-tuning. While this ensures reproducibility, it may limit the diversity and representativeness of the training data, potentially affecting the model's generalizability across broader domains. While this study compared two widely used multilingual models, exploring domain adaptation on larger or more recent architectures could offer deeper insights. Lastly, the classification performance may be influenced by text length or writing style, which are not explicitly controlled for in this study.

## Ethical Statement and Broad Impact

**Ethical Statement**   This study utilizes publicly available datasets for domain adaptation and fake news detection in Urdu. We acknowledge that the pretraining and fine-tuning data may carry inherent biases, which can be propagated or amplified by the model. Furthermore, automated systems for misinformation detection may produce false positives or negatives, especially in politically or socially sensitive contexts. We recommend that such systems be deployed with caution and always in conjunction with human oversight. This work intends to advance research in low-resource language processing, not to serve as a definitive tool for content moderation or censorship.

**Broader Impact**   This work contributes to the advancement of fake news detection for low-resource languages, with a focus on Urdu. By leveraging domain adaptation techniques and publicly available datasets, it offers a replicable framework for enhancing language models in underrepresented linguistic settings. While not intended as a standalone solution, the proposed approach has the potential to assist journalists, fact-checkers, and civil society organizations as part of broader misinformation detection workflows. The methodology is adaptable to other low-resource languages, supporting further research in multilingual NLP. We envision this work as a step toward promoting digital media literacy and fostering more resilient information ecosystems in multilingual communities.

