# OpenReview forum: "Fake News Classification in Urdu: A Domain Adaptation Approach for a Low-Resource Language"
_TMLR — Rejected by TMLR_

### Review · Reviewer_cmvs · 2026-03-12

**Summary Of Contributions:**

1. The paper studies fake news classification in Urdu, a genuinely underexplored low-resource language.
2. The evaluation spans four public Urdu fake news datasets, including both headline-only and article-level settings. This is stronger than evaluating on a single dataset and helps test whether gains are consistent across task formats.
3. The comparison between XLM-R and mBERT is valuable. One interesting empirical takeaway is that DAPT appears consistently helpful for XLM-R but less reliable for mBERT, which is practically relevant for multilingual low-resource NLP.
4. The paper does not overclaim universal gains. The fact that mBERT shows strong intrinsic improvement after DAPT but mixed downstream gains is an informative result and suggests that better language modeling does not necessarily imply better classification.

Weakness:
1. There is a major contradiction regarding the UrduFake2021 dataset. Table 3 shows mBERT (DA) gaining 11 points in F1 (0.58 to 0.69), but the text claims the domain-adapted model underperformed. This discrepancy makes it impossible to verify the paper's primary conclusions about mBERT.
2. The reported gains for XLM-R are often small (e.g., +0.01). Without reporting mean and standard deviation across multiple seeds (n≥3), it is unclear if these improvements are statistically significant or merely the result of stochastic variation in fine-tuning.
3. The claim of "Whole-Word Masking" for XLM-R is non-trivial due to the SentencePiece tokenizer. The authors do not provide sufficient evidence or detail on how word boundaries were preserved during masking, which is critical for reproducibility.

**Additional Comments:**

The two-stage fine-tuning approach is a highlight of the methodology. If the authors can resolve the reporting inconsistencies and provide statistical error bars, this will be a solid contribution to the Urdu NLP ecosystem.

**Audience:**

Yes

**Audience Explanation:**

the authors have committed to releasing their domain-adapted models, code, and dataset splits. In low-resource settings, the availability of such artifacts is often as valuable as the research findings themselves, as it provides a reproducible benchmark and a starting point for other researchers to build upon. This ensures the work will be of interest to those in Low-Resource NLP, and NLP multilingual research.

**Broader Impact Concerns:**

The paper addresses a socially beneficial task (misinformation detection). However, there is a minor risk that domain-adapting on a specific news corpus could bake in stylistic biases from certain publishers. The authors should briefly acknowledge that the "truthfulness" of the model is content-dependent and might reflect the biases of the pre-training corpus

**Claims And Evidence:**

Yes

**Claims Explanation:**

While the overall approach is sound, the evidence is currently undermined by several technical and reporting issues that must be addressed to meet TMLR's standards for correctnes, Details of my questions is stated above in the limitations

**Requested Changes:**

1. Correct the discrepancy between the text and Table 3 regarding mBERT’s performance on UrduFake2021.

2.  Re-run primary experiments over at least three random seeds and report the mean and standard deviation for all downstream metrics.

3. Provide a technical explanation of how Whole-Word Masking was implemented for Urdu using the SentencePiece tokenizer.

---

> ### Author Response · Authors · 2026-04-02
>
> We are sincerely grateful for the time and effort invested by the reviewer and appreciate the reviewer’s recognition of our work.
>
> > Q1: There is a major contradiction regarding the UrduFake2021 dataset. Table 3 shows mBERT (DA) gaining 11 points in F1 (0.58 to 0.69), but the text claims the domain-adapted model underperformed. This discrepancy makes it impossible to verify the paper's primary conclusions about mBERT.
>
> **Answer:** We thank the reviewer for pointing this out. This was an oversight on our part, which has now been corrected in the revised manuscript. The updated text is consistent with Table 3 and accurately reflects the performance of mBERT after domain adaptation.
>
> > Q2: The reported gains for XLM-R are often small (e.g., +0.01). Without reporting mean and standard deviation across multiple seeds (n≥3), it is unclear if these improvements are statistically significant or merely the result of stochastic variation in fine-tuning.
>
> **Answer:** We agree that reporting variability across runs is important, particularly when performance gains are modest. To address this, we have conducted additional experiments using five random seeds and will report the mean and standard deviation for all models and datasets in the revised manuscript. This provides a more comprehensive assessment of the observed improvements and confirms that the gains from domain adaptation, particularly for XLM-R, remain consistent across runs. We also observe an interesting pattern regarding standard deviation in classification results across long-form and short-text datasets, which will be further discussed in the revised manuscript.
>
> > Q3: The claim of "Whole-Word Masking" for XLM-R is non-trivial due to the SentencePiece tokenizer. The authors do not provide sufficient evidence or detail on how word boundaries were preserved during masking, which is critical for reproducibility.
>
> **Answer:** We thank the reviewer for this important observation. We agree that implementing whole-word masking for models using SentencePiece tokenization, such as XLM-R, requires careful handling. In our implementation, we follow the Hugging Face MLM fine-tuning procedure using the fast tokenizer interface. During tokenization, we retain the word_ids() mapping, which associates each token with its originating word in the input sequence. In the data collator, tokens sharing the same word ID are grouped together, and masking is applied jointly to all token pieces belonging to a selected word. We note that for SentencePiece-based tokenization, this approach relies on tokenizer-provided alignment and therefore represents an approximation of word-level masking rather than strictly defined linguistic word boundaries. We will release our code and clarify this implementation detail in the revised manuscript to ensure reproducibility.

---

### Review · Reviewer_65pR · 2026-03-13

**Summary Of Contributions:**

The paper studies fake news detection for Urdu by applying unsupervised domain-adaptive pretraining to multilingual PLMs (XLM-R and mBERT) on a Urdu news corpus, followed by supervised fine-tuning on four Urdu fake news datasets. Domain adaptation is implemented via MLM on Urdu news, and downstream classifiers add a small MLP. Experiments show that domain-adapted XLM-R achieves much lower perplexity than vanilla XLM-R and yields consistent though mostly modest gains in F1/accuracy across all datasets, while domain-adapted mBERT exhibits large perplexity reductions but only mixed or even negative gains on classification performance.

**Strengths**
1. The problem tackled in this paper (fake news detection on low-resource language) is of high importance in my opinion.
2. The writing is clear and easy to follow.
3. The experimentation is somewhat thorough, with four diverse Urdu fake-news datasets used.

**Weaknessess**

1. The technical novelty is very low in my opinion. The contribution is essentially a straightforward application of domain-adaptive MLM pretraining plus standard classifier heads.There is no new adaptation algorithm or modeling idea beyond existing best practices.

2. The paper observes that domain adaptation version of mBERT has much lower perplexity but does not meaningfully improve (and sometimes hurts) downstream performance, but offers only high-level speculation without deeper diagnostics.

3. All results appear to be single-run. There are no standard deviations or significance tests.

4. Finally, some broader statements about the importance of model choice and the unreliability of perplexity as a proxy for downstream performance are valid but would be more convincing with additional models or tasks.

**Audience:**

No

**Audience Explanation:**

Although the problem is important, I think the focus of this paper on a specific language makes it harder for this paper to be interesting to a larger community, as it also does not introduce methods that can be used for the general problem of fake news detection on low-resource language. Lack of technical novelty in the paper is also the reason why this paper would not likely be interesting for TMLR community. The paper uses already very-well researched methods on well-explored tasks. The weaknesses in experimental protocols would also likely make this paper less interesting.

**Claims And Evidence:**

No

**Claims Explanation:**

* Results appear to be from single runs with no standard deviations, confidence intervals, or significance tests, yet many of the reported “improvements” are just a few points in F1/accuracy. Without variability estimates, the central claims about consistent gains from domain adaptation (especially for XLM-R) are not convincingly supported.
* A key claim is that domain adaptation reduces perplexity but does not always improve downstream performance and that this effect is architecture-dependent. However, there is no deeper empirical analysis that would make this claim compelling.

**Requested Changes:**

* The paper would benefit from the reporting of means and standard deviations over at least 3–5 random seeds for all models/datasets.
* It would be nice to deeper investigate why mBERT DA lowers perplexity but yields mixed/negative downstream results. A shallow overall explanation is insufficient.
* One other change would be to test cross-dataset generalization. For example, authors can conduct leave-one-out experiments (train on 3 datasets, test on held-out one) and/or domain-shift evaluation (train on news-style datasets, test on social-media headlines) to directly validate claims about domain adaptation's robustness benefits.
* Stronger ablation design, for example ablating hyperparameter and classifier design choices would strengthen this paper.

While these changes would help strengthen the paper, the major issues regarding technical contribution would still limit the target audience. Therefore, I tend to lean towards rejection regardless.

---

> ### Author Response · Authors · 2026-04-02
>
> We are sincerely grateful for the reviewer's time and effort and for the insightful comments.
>
> > Q1: The technical novelty is very low in my opinion. The contribution is essentially a straightforward application of domain-adaptive MLM pretraining plus standard classifier heads.There is no new adaptation algorithm or modeling idea beyond existing best practices.
>
> **Answer:** We agree that the individual components used in this work, such as domain-adaptive pretraining and fine-tuning, are based on established practices. However, our contribution lies in systematically evaluating their effectiveness in the context of Urdu, a low-resource language where domain-specific data is limited, and multilingual models often underperform due to vocabulary and distribution mismatch. Instead of proposing a new modeling architecture, our goal is to provide a careful empirical analysis of how domain adaptation interacts with different multilingual models and how this translates to downstream performance across multiple datasets. Specifically, our findings highlight that improvements in intrinsic metrics such as perplexity do not consistently lead to gains in downstream tasks, and that this behavior varies across model architectures. We will revise the manuscript to more clearly position this contribution as an empirical study and to better emphasize the insights derived from these observations.
>
> > Q2: The paper observes that domain adaptation version of mBERT has much lower perplexity but does not meaningfully improve (and sometimes hurts) downstream performance, but offers only high-level speculation without deeper diagnostics.
>
> **Answer:** We agree that the behavior of mBERT, where domain adaptation substantially reduces perplexity but does not consistently improve downstream classification, requires deeper analysis.
>
> In the current manuscript, we provide a high-level explanation, attributing this behavior to differences in model characteristics, including XLM-R’s larger model size, improved Urdu coverage in pretraining, and the use of dynamic masking. We acknowledge that this discussion is brief and will expand it in the revised version. Specifically, we will more clearly describe architectural and training differences between the models, including dynamic masking in XLM-R versus static masking in mBERT, and the absence of the next sentence prediction (NSP) objective in XLM-R.
>
> In addition, we conducted further analysis to investigate whether the observed behavior may be related to overfitting during domain-adaptive pretraining. Specifically, we repeated the domain adaptation for mBERT using a reduced subset of the corpus (500k samples instead of 1M), which resulted in a slightly higher perplexity (2.43 vs. 2.10). Interestingly, this model yielded improved downstream classification performance (measured over multiple random seeds), suggesting that, for mBERT, aggressively optimizing the MLM objective may lead to overfitting to the pretraining task, which does not consistently translate to improved classification performance.
>
> We will include this additional experiment in the revised manuscript to provide a more detailed analysis of the relationship between perplexity and downstream performance for mBERT.
>
> > Q3: All results appear to be single-run. There are no standard deviations or significance tests.
>
> **Answer:** We have now conducted additional experiments using five random seeds and will report the mean and standard deviation for all models and datasets in the revised manuscript. This provides a more comprehensive assessment of the observed improvements and confirms that the gains from domain adaptation, particularly for XLM-R, remain consistent across runs. We also observe an interesting pattern in the standard deviation of classification results across long-form and short-text datasets, which we will further discuss in the revised manuscript.
>
> > Q4: Finally, some broader statements about the importance of model choice and the unreliability of perplexity as a proxy for downstream performance are valid but would be more convincing with additional models or tasks.
>
> **Answer:** We agree that broader validation across additional models or tasks could further strengthen the generality of the findings. In this work, our goal is to provide a controlled analysis of domain adaptation within a low-resource Urdu setting using widely adopted multilingual models. Expanding the study to include additional architectures or task settings would introduce further variability in model capacity and pretraining conditions, making it more difficult to isolate the effect of domain adaptation. We will clarify this scope more explicitly in the revised manuscript and position our findings as an initial empirical analysis in a low-resource setting, which can be extended in future work.

---

> > ### Author Response · Authors · 2026-04-02
> >
> > >Q5: A key claim is that domain adaptation reduces perplexity but does not always improve downstream performance and that this effect is architecture-dependent. However, there is no deeper empirical analysis that would make this claim compelling.
> >
> > **Answer:** In our work, we do not intend to make a broad general claim, but rather report an empirical observation based on two widely used multilingual models (XLM-R and mBERT). Across multiple datasets and runs, we consistently observe that domain adaptation improves downstream performance for XLM-R, while for mBERT it leads to substantial reductions in perplexity but mixed downstream results. To further investigate this behavior, we conducted additional experiments (reported in the revision), including multi-seed evaluation and controlled variation in domain adaptation data size. These analyses reinforce the observed discrepancy between intrinsic and downstream performance for mBERT. We will clarify in the revised manuscript that our findings are specific to the models and setting considered.
> >
> > >Q6: Although the problem is important, I think the focus of this paper on a specific language makes it harder for this paper to be interesting to a larger community, as it also does not introduce methods that can be used for the general problem of fake news detection on low-resource language. Lack of technical novelty in the paper is also the reason why this paper would not likely be interesting for TMLR community. The paper uses already very-well researched methods on well-explored tasks. The weaknesses in experimental protocols would also likely make this paper less interesting.
> >
> > **Answer:** Urdu is spoken by over 200 million people around the world and represents a genuinely underexplored setting in NLP, with increasing research interest in low-resource languages. We therefore believe this work would be of interest to researchers working in low-resource settings in general, and Urdu in particular. Secondly, we view Urdu as a representative case study for low-resource scenarios, where domain-specific data is limited, and multilingual models often underperform due to vocabulary and distribution mismatch. The main objective of this work is to investigate whether domain-adaptive pretraining can help bridge this gap in the absence of strong language-specific models. As such, the methodology and insights presented here are not specific to Urdu and can be extended to other low-resource languages.
> >
> > Regarding technical novelty, we agree that the individual components are based on established methods. However, our contribution lies in providing a systematic empirical analysis of how domain adaptation interacts with different multilingual models and how this translates to downstream performance across multiple datasets. Specifically, our findings highlight that improvements in intrinsic metrics such as perplexity do not consistently translate to downstream gains, and that this behavior varies across model architectures.
> >
> > We will revise the manuscript to better emphasize both the broader applicability of the approach and the empirical insights derived from this study.

---

### Review · Reviewer_SJ1S · 2026-03-25

**Summary Of Contributions:**

The paper tackles fake news classification in Urdu and investigates whether the domain-adaptive pretraining can help PLMs to solve this task. The author proposes domain adaptation via masked language modeling on a 1M Urdu news corpus before fine-tuning it for binary classification on four Urdu fake news datasets. The results show that domain-adapted XLM-R consistently outperforms its vanilla version across all datasets. In contrast, while mBERT shows significantly reduced perplexity after adaptation, its downstream classification gains remain inconsistent.

**Additional Comments:**

N/A

**Audience:**

Yes

**Audience Explanation:**

Strength

1.The motivation is clear. The paper addresses a meaningful and underexplored setting: fake-news detection for Urdu, where both domain datasets and pretrained models are limited.

Weaknesses

1. The technical approach is not specific for Urdu. The author primarily applies existing domain-adaptive pretraining and fine-tuning techniques to Urdu fake-news detection. While applying them to Urdu is valuable, the paper does not highlight the unique challenges of Urdu in multilingual misinformation settings, which weakens the broader research contribution.

**Claims And Evidence:**

No

**Claims Explanation:**

Strengths


1. The experimental design is comprehensive and easy to follow. The authors evaluate their approach across four diverse datasets encompassing both short headlines and long articles.

2. The importance of domain-specific adaptation is well-supported. The study provides a transparent comparison between the vanilla model and the adapted model. The results of perplexity scores reveal that domain-specific fine-tuning is essential for the detection performance in Urdu.

3. The two-stage fine-tuning strategy is effective. The author first freezes the base model initially to prevent catastrophic forgetting before unfreezing with a lower learning rate. The strategy is supported by the training stability and convergence observations detailed in Section 4.1.2.

Weaknesses

1. The analysis of mBERT is insufficient. Although domain adaptation substantially reduces mBERT perplexity, it does not consistently improve the fake-news detection performance. However, the paper does not investigate why this result occurs. It remains unclear whether the issue is related to the BERT architecture itself or the mBERT’s pretraining data distribution. A stronger analysis could include additional BERT-style baselines like IndicBERT [1] or other Urdu-specific BERT variants to isolate the issue.

2. The advantage of domain adaptation on detection performance appears modest. Although domain adaptation clearly improves perplexity, the gains over vanilla fine-tuning are generally small (e.g., 0.84 vs 0.85 on UFAKE21 in Table 3). To assess statistical robustness, the authors could report results from multiple runs or provide confidence intervals.

3. The performance degradation on long-form articles highlights the limitation of the proposed architecture. The author notes that results on UrduFake2021 are worse than on headline-based datasets, which also reflects a limitation of BERT-style models on long documents. Truncation may discard important evidence since realistic fake-news detection often requires full-article reasoning rather than headline classification. The limitation reduces the practical applicability of the proposed approach.

4. The evaluation lacks comparisons with recent autoregressive LLMs (e.g. LLaMA or Mistral series) that support Urdu. Given their scale and zero-shot capabilities, they may serve as stronger baselines and better contextualize the competitiveness of the proposed encoder-based approach.

[1] Dabre R, Shrotriya H, Kunchukuttan A, et al. IndicBART: A pre-trained model for indic natural language generation[C]//Findings of the Association for Computational Linguistics: ACL 2022. 2022: 1849-1863.

**Requested Changes:**

See the weaknesses described in the answers.

---

> ### Author Response · Authors · 2026-04-02
>
> We are sincerely grateful for the time and effort invested by the reviewer and the insightful comments.
>
> > Q1:  The analysis of mBERT is insufficient. Although domain adaptation substantially reduces mBERT perplexity, it does not consistently improve the fake-news detection performance. However, the paper does not investigate why this result occurs. It remains unclear whether the issue is related to the BERT architecture itself or the mBERT’s pretraining data distribution. A stronger analysis could include additional BERT-style baselines like IndicBERT [1] or other Urdu-specific BERT variants to isolate the issue.
>
> **Answer:** We agree that the behavior of mBERT, where domain adaptation substantially reduces perplexity but does not consistently improve downstream classification, requires deeper analysis.
>
> In the current manuscript, we provide a high-level explanation, attributing this behavior to differences in model characteristics, including XLM-R’s larger model size, improved Urdu coverage in pretraining, and the use of dynamic masking. We acknowledge that this discussion is brief and will expand it in the revised version. Specifically, we will more clearly describe architectural and training differences between the models, including dynamic masking in XLM-R versus static masking in mBERT, and the absence of the next sentence prediction (NSP) objective in XLM-R.
>
> In addition, we conducted further analysis to investigate whether the observed behavior may be related to overfitting during domain-adaptive pretraining. Specifically, we repeated the domain adaptation for mBERT using a reduced subset of the corpus (500k samples instead of 1M), which resulted in a slightly higher perplexity (2.43 vs. 2.10). Interestingly, this model yielded improved downstream classification performance (measured over multiple random seeds), suggesting that, for mBERT, aggressively optimizing the MLM objective may lead to overfitting to the pretraining task, which does not consistently translate to improved classification performance.
>
> We will include this additional experiment in the revised manuscript to provide a more detailed analysis of the relationship between perplexity and downstream performance for mBERT.
>
> > Q2: The advantage of domain adaptation on detection performance appears modest. Although domain adaptation clearly improves perplexity, the gains over vanilla fine-tuning are generally small (e.g., 0.84 vs 0.85 on UFAKE21 in Table 3). To assess statistical robustness, the authors could report results from multiple runs or provide confidence intervals.
>
> **Answer:** We agree that reporting variability across runs is important, particularly when performance differences are modest. To address this, we have conducted additional experiments using five random seeds and will report the mean and standard deviation for all models and datasets in the revised manuscript. This provides a more comprehensive assessment of the observed improvements and confirms that the gains from domain adaptation, particularly for XLM-R, remain consistent across runs. We also observe an interesting pattern regarding standard deviation in classification results across long-form and short-text datasets, which will be further discussed in the revised manuscript.
>
> > Q3:  The performance degradation on long-form articles highlights the limitation of the proposed architecture. The author notes that results on UrduFake2021 are worse than on headline-based datasets, which also reflects a limitation of BERT-style models on long documents. Truncation may discard important evidence since realistic fake-news detection often requires full-article reasoning rather than headline classification. The limitation reduces the practical applicability of the proposed approach.
>
> **Answer:** We selected different sequence lengths for shorter and longer datasets based on average text length, with the aim of retaining as much information as possible. We will further clarify the sequence length selection process in the revised manuscript, but we agree that BERT-style models, due to fixed input length constraints, may lose information from longer examples due to truncation, which can impact performance on datasets such as UrduFake2021. In addition, UrduFake2021 is the smallest dataset among those considered, which may further contribute to its weaker performance. More broadly, predicting the veracity of long-form articles is inherently more challenging than short-text classification, as such articles may contain a mixture of factual and misleading content, making binary labeling less straightforward for the model. We will clarify this limitation in the revised manuscript.

---

> > ### Author Response · Authors · 2026-04-02
> >
> > > Q4: The evaluation lacks comparisons with recent autoregressive LLMs (e.g. LLaMA or Mistral series) that support Urdu. Given their scale and zero-shot capabilities, they may serve as stronger baselines and better contextualize the competitiveness of the proposed encoder-based approach.
> >
> > **Answer:** While recent autoregressive LLMs such as LLaMA and Mistral demonstrate strong zero-shot capabilities, our study focuses on evaluating the impact of domain-adaptive pretraining within a controlled encoder-based setting. Comparing with large autoregressive models would introduce differences in model scale, architecture, and pretraining data, making it difficult to isolate the effect of domain adaptation. Our goal is to provide a controlled comparison between vanilla and domain-adapted variants of the same model. We will clarify this experimental scope more explicitly in the revised manuscript.
> >
> > > Q5: Technical approach not specific to Urdu
> >
> > **Answer:** We agree that individual components such as domain adaptation and finetuning are general but our contribution lies in systematically evaluating the impact of domain adaptation before finetuning in the context of Urdu, a low-resource language where domain-specific data is limited and multilingual models often underperform due to vocabulary and distribution mismatch.  Our study highlights how domain adaptation interacts with different multilingual models in Urdu and affects downstream performance. Specifically, we investigate whether domain adaptation can help bridge the gap in the absence of strong language-specific models for low-resource settings. We will clarify this rationale more clearly in the revised manuscript

---

### Decision · Action_Editor_k8yY · 2026-05-14

**Recommendation:** Reject

**Additional Comments:**

The manuscript was reviewed by three experts. After going through the author's response carefully, two of the reviewers lean towards rejection of the manuscript.

Though the problem addressed is important, there are several issues with the current version. The primary concern is that it will not be interesting to a wider audience since the application is only on one language and it is not clear whether the observations generalize to other low-resource languages. In addition, the behaviour of the models are not analyzed satisfactorily and it is not clear whether these observations will hold for other models as well, making the study quite limited in scope. Although not the main concern used for making the recommendation, the reviewers also noted the limited novelty of the work.

Thus the recommendation is to reject the paper in its current form.

**Audience:**

No

**Audience Explanation:**

This paper addresses an important problem. But since only one language is considered, it may not be interesting to the TMLR audience in general. Also, it is not clear whether the observations are generalizable to other low-resource languages.

**Claims And Evidence:**

No

**Claims Explanation:**

In this work, the authors study the effectiveness of domain adaptation before fine-tuning two multilingual models for fake news classification in Urdu. The observations are limited to these two models and it is not clear whether it extends to other models as well. Also, it is not clear whether this study extends to other low-resource languages other than Urdu.

**Resubmission Of Major Revision:**

The authors may consider submitting a major revision at a later time.